# It's not who you know, but who you are: Explaining income gaps of stigmatized-caste business owners in India

**Prateek Raj[1], Thomas J. Roulet[2]\*, Hari Bapuji[3]**

**1** Department of Strategy, IIM Bangalore, Bangalore, Karnataka, India, **2** Judge Business School and King's College, University of Cambridge, Cambridge, Cambridgeshire, United Kingdom, **3** Department of Management and Marketing, University of Melbourne, Melbourne, Victoria, Australia

\* t.roulet@jbs.cam.ac.uk

**Data Availability Statement:** IHDS is a nationally representative survey that is a collaborative project between three universities in the U.S. (University of Maryland, Indiana University and the University of Michigan) and the National Council of Applied

## Abstract

Scholars across disciplines and around the world have diverted research attention to rising income inequalities across groups and strategies to reduce them. The literature has broadly identified human capital and social capital as two potential tools to facilitate economic mobility and to reduce inequalities. However, it is not known whether these tools work equally well for stigmatized groups, particularly in societies with systemic inequalities. Analyzing data from a pan-India survey, we show that business owners from stigmatized groups (i.e., Dalits in India, who are stigmatized as untouchables) experience a business income gap of around 16% compared to others, including those business owners who are from communities that are disadvantaged but are not similarly stigmatized. We find that, instead of being reduced, this gap in fact increases at higher levels of social capital, especially bridging social capital, illustrating the social processes of stigmatization that limit the benefits that Dalits can reap from social capital. By contrast, Dalits can reap similar income benefits as others from human capital. Our results show that human capital helps stigmatized groups mitigate the implications of stigma, but social capital does not.

## Introduction

The World Inequality Report 2022 reported that the top 10% of earners took home 52% of the global income, while the bottom 50% earned just 8% of the total income. Understanding the antecedents of such stark income inequalities is a major challenge of our times. Systematic and granular studies documenting inequality generating processes are important to grasp this complex issue. This study addresses the issue of income disadvantages faced by business owners from stigmatized castes in India. Despite the policy focus on reducing income disparities through human and social capital, it remains unclear whether these strategies are equally effective for marginalized groups in societies ridden with systemic inequalities. This study aims to fill this gap by analyzing data from a nationwide survey in India to quantify the income disparity faced by Dalit business owners (who are stigmatized as untouchables) compared to other disadvantaged but non-stigmatized groups.

Economic Research (NCAER) of India. It is available to download here: https://ihds.umd.edu/data/data-download.

**Funding:** The author(s) received no specific funding for this work.

**Competing interests:** The authors declare that there is no competing interest exists.

As the study of inequality and its consequences becomes a major academic focus [1, 2], management scholarship has also renewed its attention on inequality [3–6]. One of the key phenomena that has been connected in the literature to inequality is stigma [7, 8]. However, most of the existing work views stigma as dyadic and examines its isolated consequences for the stigmatized [7, 9, 10]. Instead, there are benefits in looking more holistically at the diffused nature of prejudice in some cases of widespread discrimination, in which the interrelated processes interact in social contexts ultimately assigning lower worth to a stigmatized group and limiting their access to resources, thus generating inequality [10–13].

To explain how diffused prejudice in social contexts disadvantages groups, we conceptualize such stigma as institutional stigma, i.e., stigma that is ascribed to individuals based on their membership in a demographic group [14], but operates in a diffused manner through multiple interrelated social processes [11], and reflects societal power structures and systemic oppression of certain groups [15]. In this sense, the prejudice incurred by stigma is carried through multiple "institutionalized" channels [16]. These include access to resources and opportunities, as well as dignity as an individual [17]. Other non-stigmatized groups will not face such hurdles related to stigma. Accordingly, we argue that business owners from stigmatized groups (e.g. Dalits), with similar attributes as those from other disadvantaged groups, will have lower income. We further study how social and human capital interact with institutional stigma, influencing income outcomes. By examining these dynamics, our research provides valuable insights for developing more effective policies and strategies to mitigate economic inequalities experienced by stigmatized communities.

## Literature review

Economic inequalities in a society refer to "uneven distribution in the endowment and/or access to financial and non-financial resources in a society, which manifests in differential abilities and opportunities to engage in value creation, appropriation, and distribution" [18]. Such unevenness often manifest along demographic lines, such as gender, race, and caste [13, 17, 19] and get reflected in economic structures and employment relations, conferring income advantages to privileged demographic groups, while at the same time disadvantaging marginalized groups [20]. For example, women face discrimination in hiring levels and wages resulting in losses of "a difference of half a job level and 15% in wages [21]. Similar income disparities exist for racial minorities [22], LGBT employees [23], and employees from disadvantaged social backgrounds [24, 25].

While the reproduction of inequalities due to discrimination is well-researched [20, 26], less attention has been given to how stigma affects inequality in society. Stigma is often conceptualized as dyadic [7, 9, 10], but this perspective overlooks broader questions about the origins and purposes of stigma as a tool of social control by dominant groups [15].

Stigma need not be operational in dyadic interactions alone, but can also be widespread and culturally anchored [27] as well as structurally diffused [14, 28]. This "institutional stigma" [29] involves negative public perceptions about (public stigma) and adverse expectations from (structural stigma) social groups [11] who get devalued in a particular social context [30]. Institutional stigma marginalizes groups viewed as deviant by privileged groups, serving as a tool of social control [16].

The caste system, the way in which it stigmatizes members especially of the lower caste Dalit community, is an example of institutional stigma. The caste system places individuals on a graded hierarchy based on birth, and assigns them different occupational roles and social statuses, governing their social and economic life [31]. Among the different caste groups, the members of the lowest castes in India, Dalits, are particularly stigmatized facing denial of

access to education, civic facilities, public spaces, markets, and places of worship [17, 32, 33], as well as business opportunities [31, 34, 35]. This stigma is rooted in the public devaluation of their worth and structural pressures to marginalize them [11, 12, 36].

Institutional stigma of caste is widespread, diffuse, and multifaceted, without a specific visible marker like skin colour, ascertained via a range of social cues and processes such as last names, food habits, and traditional family occupation, and cultural interests [37]. Such cues can enable stigma, often without the stigmatizing agents consciously realizing these, causing material disadvantages to those who are stigmatized through two mechanisms—evaluation of stigmatized groups and resource distribution [12]. Stigmatized groups face recognition gaps, affecting their access to resources and opportunities, leading to a recursive loop where limited opportunities exacerbate the stigma [38]. Stigma excludes the stigmatized from community cores, hampering relationships and access to resources [13, 39, 40].

## Methodology

For business owners from stigmatized groups, stigma results in lower recognition as potential partners and reduced access to competitive resources [13]. Institutional stigma's public and structural nature perpetuates these disadvantages [11]. Members of groups that suffer institutional stigma experience the outright denial of opportunity (e.g., new consumers and suppliers will avoid them) and higher cost of opportunities (e.g., more investment and effort to secure consumers or contracts). Consequently, all other sources of business (dis)advantage being equal, we expect institutionally stigmatized business owners (Dalits) to generate lower income from their businesses compared to similar business owners from other castes, including those who are marginalized due to caste or religion.

**Hypothesis 1 (H1):** Business owners from stigmatized groups (i.e., Dalits) have lower income compared to business owners from other disadvantaged groups, all else being equal.

Given the diffused and social nature of institutional stigma and the different extents to which social capital and human capital [41–45] rely on social processes, we suggest that latter have differential effects on the relationship between stigma and income disadvantage.

In the context of India, while non-stigmatized groups may benefit from their social capital, Dalit business owners will likely receive lower returns for their social capital due to frequent rejections, micro-aggressions, and the burden of managing their stigmatized identity, which hinders their ability to secure valuable business opportunities [46, 47]. We posit that the disadvantages due to institutional stigma are accentuated at higher levels of social capital because institutional stigma operates in a cultural and normative way, limiting people from closely engaging with those who are stigmatized, even in the presence of social relations between them [13, 17, 48], thereby keeping those who are stigmatized from generating returns from social capital to the same degree that non-stigmatized can.

**Hypothesis 2 (H2):** The income disadvantage for stigmatized business owners (i.e., Dalits), as compared to non-stigmatized business owners, is greater at higher social capital.

Social capital can be of two types—bonding and bridging social capital [49]. Bonding social capital relates to connections within one's own community, friends, and family, cementing such homogenous groups; whereas bridging social capital is about beyond-community ties, that connect one to different, diverse, communities. We argue that the disadvantages due to institutional stigma, being a result of out-group prejudice, is likelier to be accentuated at higher levels of bridging social capital.

**Hypothesis 3 (H3):** The income disadvantage for stigmatized business owners (i.e., Dalits), as compared to non-stigmatized business owners, is greater at higher levels of bridging social capital.

In contrast to social capital, human capital captures individuals' capabilities, rather than social relationships [42], and thereby we expect it to attenuate income disadvantages due to institutional stigma.

**Hypothesis 4 (H4):** The income disadvantage for stigmatized business owners (i.e., Dalits), as compared to non-stigmatized business owners, is lower at higher levels of human capital.

Biographical accounts have highlighted the importance of social and human capital for business owners from Dalit communities. For example, Kapur, Babu and Prasad [50], emphasizing the role of both an entrepreneur's immediate social capital and human capital, write in their biographic accounts of successful Dalit entrepreneurs;

"*As one reads the stories in this book, some commonalities emerge in the routes to success for these Dalits. Often, their success was nurtured by one or both parents (and especially the mother) who saw the benefit of providing their child with an education. They ensured funds for the child's education by taking on extra backbreaking work that provided the ready cash, sometimes choosing between their children when finances were severely strained and allowed the education of only one child. Often the siblings would also make sacrifices by taking less so that the pooled resources would be above a minimum threshold. This allowed a chosen child to access opportunities that would create a pathway to success for one member, who in time would hopefully 'pull up' the entire family.*

*A common thread among these stories is the jugaad employed by these budding entrepreneurs to find start-up funds for their new ventures. These funds are often provided by family and friends-the wife selling her jewellery, the mother relinquishing her savings, and other family members and well-wishers chipping in. Without this initial capital, it is unlikely that these fledgling entrepreneurs would have made a success of their ventures. . .The near absence of linkages with a financial system dominated by public sector banks, supposedly enjoined to lend for 'priority sectors', matters even more since Dalit entrepreneurs lack access to the sorts of social networks that are now recognized as critical for success in any walk of life, especially entrepreneurship.*"

We take a quantitative approach in our study, in the empirically rich and diverse [51] context of India, where the caste system operates as a socially constructed multidimensional institution that creates inequalities by constructing differences among people [52]. Scholars suggest that "people are born into their caste identity. It is in their skin, their name, their relationships, their work and their habits, and it is constantly reinforced by others in social interactions" [53]. Within the graded hierarchy of caste, Dalits (also known as untouchables) are placed at the bottom and face a stigma that is different from other types of stigmas (e.g., stigma due to ethnicity, occupation, or deviant behavior) because they are seen as polluted and impure individuals, and are dehumanized [10, 17, 53]. This stigma, we argue is institutional stigma that creates income gaps for Dalits, but not for others, including those who may otherwise be disadvantaged due to caste system.

To test our hypotheses on the direct effect of institutional stigma on business income and the moderating effects of social capital and human capital, we need a dataset that includes these data as well as data on a wide array of business owner, business and geographic characteristics to control for potential confounds. In addition, we also need to be able to establish the

influence of stigma net of broader socioeconomic disadvantages faced by other groups, but are not institutionally stigmatized. Accordingly, we used the India Human Development Survey (IHDS) of 2011, which is publicly available data [54]. IHDS is a nationally representative survey that is a collaborative project between three universities in the U.S. (University of Maryland, Indiana University and the University of Michigan) and the National Council of Applied Economic Research (NCAER) of India.

The IHDS surveys over 42,000 households in different demographic groups, in 373 districts across India, and provides a detailed picture of a pan-Indian sample of households. The survey records a household's caste category, as well as the more granular caste name (which is necessary to tease out the income disadvantage due to institutional stigma over and above the disadvantage due to belonging to other non-privileged caste groups), whether they own a business and their income, along with an array of other questions relevant for our study. Of the households surveyed, 21 percent (around 8,800) own at least one non-farm business. These are micro and small businesses, with an average annual income of around INR 100,000 (i.e., approximately US$ 1,350), from 67 classified industries. We use the IHDS data to construct our study variables, which we describe below.

## Dependent variable

Business income: We operationalize our dependent variable as a household's combined business income. Even though a variety of measures are used in estimating business performance [55], in cross-sectional settings, business returns such as sales [e.g., 56] and income [e.g., 57] are common. As is typical in studies of small businesses [e.g., 58], the distribution of data on this variable is highly skewed in our dataset, and we log transform it to get a more even distribution.

## Independent variables

Institutional stigma (Dalit): We capture institutional stigma with a dummy variable (1, if the household consists of Dalits, and 0 otherwise). Households that belong to a group of castes (listed as Scheduled Castes and officially referred to as such by the government) that were socially stigmatized, and were considered "untouchables" belong to this Dalit group. This information is self-reported in the IHDS survey, which is the standard way in which all surveys and censuses on caste identity have been carried in India.

Social capital: As nationally representative surveys rely on random sampling for data collection, they typically use position generators to measure social capital [59]. The IHDS survey asks households about their social capital and reports their acquaintance with members of 10 different professions (i.e., elected officers, local politicians, government employees, other officers, inspectors, other police, doctors, health workers, teachers, and school workers). We count the number of professions (the categories above) across which a household reported as having a personal acquaintance and use this to proxy for the size of their social capital. Specifically, a household might have no personal acquaintance with a member of any of these professions at all (i.e., a 0 for both within and outside the community), or it can have a connection with a member of each of the 10 professions, and furthermore have these connections to contacts who are within their community as well to those who are outside. Therefore, a household can have a minimum score of 0 and a maximum score of 20.

Bridging social capital. To measure bridging social capital, we count the presence of a personal acquaintance in each of the 10 professions, but only if the personal acquaintances are outside community. Therefore, this measure could vary from 0 to 10. In testing our prediction

for bridging social capital, we also control for Bonding social capital, which is its converse, and is operationalized by using acquaintances who are within one's community.

Human capital: Education level is a commonly used proxy to capture human capital [e.g., 60, 61]. The IHDS data reports the highest adult education in the household, which varies from no schooling to above Bachelors (captured in a scale ranging from 0 to 16). When we use it as a control, we use a set of indicator variables to capture the effects for each level of Human capital separately, and when we test H4, with an interaction variable, we remove these fixed effects and operationalize it as a variable that ranges from 0 to 16, as we note above.

We control for a set of variables at the household, village/neighborhood, and demographic group level that may influence the relationships we study, which we detail further in the supplementary text. In addition to the control variables, we account for Father's Occupation, District, and Industry using a set of indicators or dummy variables, i.e., fixed effects. We use Ordinary Least Squares (OLS) regression models with robust standard errors for our estimations. Our estimation sample includes 8,506 business-owning households.

## Results

Our main results are given in Table 1. Model 1 is the baseline model, with all the control variables included. We see that social capital is a significant predictor of income (p < .001), with one standard deviation increase in social capital increasing the business income by 16%. Belonging to a disadvantaged background (Socioeconomic disadvantage) has a negative and significant (p < .01) association with business income, reducing it by 8.8%. As expected, urban location, average income of location, fewer Dalits in location, also have a positive relationship with business income, as do the number of businesses and size of the household. Untouchability experience of household does not have any effect on business income, once the other factors in our models are accounted for.

In Model 2, we add Institutional Stigma (Dalit), which has a negative and significant (p < .01) coefficient, supporting H1. All else being equal, business income of a Dalit household is 15.9% lower, compared with other households that are similar, but which do not face institutional stigma. When accounting for institutional stigma, the coefficient on Socioeconomic disadvantage variable in Model 2 is small and statistically insignificant, showing that belonging to a disadvantaged group, other than being a Dalit, is not associated with a significant disadvantage. In Model 3, we add the interaction between institutional stigma and social capital. As expected, the interaction has a negative and significant (p < .05) coefficient, supporting H2. Based on Model 3, one standard deviation increase in social capital is associated with a 17.3% increase in business income for households from non-stigmatized communities, but a similar increase in social capital is associated with only a 6% increase in business income for Dalit households. As these numbers indicate, the business income disadvantages that Dalits experience, as compared to non-stigmatized communities, are amplified at higher levels of social capital.

In Models 4–6, we decompose social capital into bridging and bonding social capital. Model 4 replicates Model 2, but with these two social capital variables instead of the aggregate social capital variable. Institutional stigma continues to have a similar coefficient as in Model 2, and both bridging and bonding social capital are positively and significantly (p < .001) associated with business income.

In Model 5, we add the interaction between institutional stigma and bridging social capital, to test H3. As hypothesized, the interaction has a negative and significant (p < .05) coefficient. According to Model 5, one standard deviation increase in bridging social capital is associated with a 10.8% increase in business income for households from non-stigmatized communities,

**Table 1. OLS estimations of the relationship between institutional stigma and business income.**

| | Model 1 | Model 2 | Model 3 | Model 4 | Model 5 | Model 6 | Model 7 | Model 8 | Model 9 |
|---|---|---|---|---|---|---|---|---|---|
| | *Baseline* | *H1 Dalit* | *H2 Social Capital* | *H3 Bridging Social Capital* | | | *H4 Human Capital* | *H2 & H4* | *H3 & H4* |
| Institutional Stigma (Dalit) | | -.173** (.057) | -.079 (.069) | -.173** (.057) | -.073 (.066) | -.073 (.068) | -.117 (.102) | -.075 (.104) | -.063 (.104) |
| Institutional Stigma X Social capital | | | -.024* (.012) | | | | | -.025* (.012) | |
| Institutional Stigma X Bridging social capital | | | | | -.041* (.017) | -.041* (.019) | | | -.042* (.019) |
| Institutional Stigma X Bonding social capital | | | | | | -.0003 (.026) | | | .0001 (.026) |
| Institutional Stigma X Human capital | | | | | | | -.006 (.009) | -4.21e-05 (.009) | -.001 (.009) |
| Social capital | .035*** (.004) | .035*** (.004) | .038*** (.004) | | | | .036*** (.004) | .039*** (.004) | |
| Bridging social capital | | | | .034*** (.008) | .039*** (.008) | .039*** (.008) | | | .039*** (.008) |
| Bonding social capital | | | | .037*** (.008) | .037*** (.008) | .037*** (.009) | | | .038*** (.009) |
| Human capital | | | | | | | .033*** (.004) | .033*** (.004) | .033*** (.004) |
| CONTROLS | | | | | | | | | |
| Untouchability experience—Household | -.171 (.116) | -.142 (.115) | -.158 (.115) | -.142 (.115) | -.156 (.114) | -.156 (.115) | -.147 (.115) | -.156 (.115) | -.154 (.115) |
| Untouchability experience—Caste (avg.) | -.263 (.155) | -.047 (.169) | -.061 (.167) | -.047 (.168) | -.058 (.167) | -.058 (.166) | -.061 (.169) | -.072 (.167) | -.069 (.167) |
| Socioeconomic disadvantage | -.093** (.034) | -.067 (.035) | -.062 (.035) | -.067 (.035) | -.063 (.035) | -.063 (.035) | -.068 (.035) | -.065 (.035) | -.066 (.035) |
| Tenure of residence | -.001 (.001) | -.001 (.001) | -.001 (.001) | -.001 (.001) | -.001 (.001) | -.001 (.001) | -.001 (.001) | -.001 (.001) | -.001 (.001) |
| Urban location | .197*** (.050) | .199*** (.050) | .201*** (.050) | .199*** (.050) | .199*** (.050) | .199*** (.050) | .196*** (.050) | .198*** (.050) | .197*** (.050) |
| Fraction of Dalits in location | -.315*** (.077) | -.217* (.088) | -.215* (.088) | -.217* (.088) | -.217* (.088) | -.217* (.089) | -.223* (.089) | -.221* (.089) | -.223* (.090) |
| Average income—Location | .341*** (.032) | .344*** (.032) | .344*** (.032) | .344*** (.032) | .344*** (.032) | .344*** (.032) | .346*** (.032) | .346*** (.032) | .347*** (.032) |
| Land | -.024 (.038) | -.028 (.038) | -.027 (.038) | -.028 (.038) | -.027 (.038) | -.027 (.038) | -.030 (.038) | -.029 (.038) | -.030 (.038) |
| Number of businesses in household | .587*** (.033) | .586*** (.033) | .584*** (.033) | .586*** (.033) | .585*** (.033) | .585*** (.033) | .587*** (.033) | .586*** (.033) | .587*** (.033) |
| Household size | .040*** (.006) | .040*** (.006) | .040*** (.006) | .040*** (.006) | .040*** (.006) | .040*** (.006) | .040*** (.006) | .040*** (.006) | .040*** (.006) |
| Human capital fixed effects | Yes | Yes | Yes | Yes | Yes | Yes | No | No | No |
| Industry fixed effects | Yes | Yes | Yes | Yes | Yes | Yes | Yes | Yes | Yes |
| Father's occupation fixed effects | Yes | Yes | Yes | Yes | Yes | Yes | Yes | Yes | Yes |
| District fixed effects | Yes | Yes | Yes | Yes | Yes | Yes | Yes | Yes | Yes |
| Constant | 6.531*** (.515) | 6.482*** (.516) | 6.466*** (.515) | 6.485*** (.516) | 6.464*** (.516) | 6.464*** (.516) | 6.465*** (.511) | 6.457*** (.510) | 6.455*** (.511) |
| Observations | 8,506 | 8,506 | 8,506 | 8,506 | 8,506 | 8,506 | 8,506 | 8,506 | 8,506 |

*(Continued)*

**Table 1.** (Continued)

| | Model 1 | Model 2 | Model 3 | Model 4 | Model 5 | Model 6 | Model 7 | Model 8 | Model 9 |
|---|---|---|---|---|---|---|---|---|---|
| | *Baseline* | *H1 Dalit* | *H2 Social Capital* | *H3 Bridging Social Capital* | | | *H4 Human Capital* | *H2 & H4* | *H3 & H4* |
| *R*-squared | .354 | .355 | .356 | .355 | .356 | .356 | .354 | .354 | .354 |

*Notes*.

• Model 2 reports the regression between Institutional Stigma (independent variable) and business income (dependent variable). Model 3 interacts Institutional Stigma dummy with Social capital. Model 4 is Model 2 with Social capital decomposed into Bridging social capital and Bonding social capital. Model 5 interacts Institutional Stigma dummy with Bridging social capital. Model 7 interacts Institutional Stigma dummy with Human capital. Model 8 interacts Institutional Stigma dummy with Social and Human capital jointly, and Model 9 interacts Institutional Stigma dummy with Bridging social and Human capital jointly. All regressions include control variables related to untouchability, socioeconomic disadvantage, and locational and household characteristics, and Father's Occupation, District, and Industry fixed effects.

• The Human capital fixed effects (measured as highest adult education in the household) are included in Models 1–6, but removed in Models 7–9, because in those models H4 is tested by interacting a single Human capital variable (varying from 0 to 16 capturing the highest adult education) interacted with stigma.

• Robust standard errors in parentheses.*** $p < .001$, ** $p < .01$, * $p < .05$ (two-tailed tests).

while a similar increase in bridging social capital is associated with no increase at all in the business income for Dalit households.

In Model 6, we interact institutional stigma with both bridging and bonding social capital. The estimate for interaction between institutional stigma and bridging social capital remains positive and significant ($p < .05$) as in Model 5, consistent with H3. The interaction between institutional stigma and bonding social capital is small and not significant ($p = .99$). According to Model 6, one standard deviation increase in bonding social capital is associated with an 8.4% higher business income for business owners from both stigmatized and non-stigmatized communities. As for the case with social capital more generally, these numbers show that the existing disadvantages that Dalits experience with respect to their business income, as compared to non-stigmatized communities, are amplified at higher levels of bridging social capital.

In Model 7, we interact institutional stigma with human capital, to test H3. In this model, we remove the human capital indicator variables from the model, and add it instead as a continuous variable (which varies from 0 to 16). The interaction variable does not have a significant coefficient ($p = .49$). One standard deviation increase in human capital is associated with 16.6% increase in business income for business owners from both stigmatized and non-stigmatized communities. This similarity in the effect size does not support H4, i.e., human capital is associated with higher business income for business owners from both stigmatized and non-stigmatized communities, and it does not attenuate (nor accentuate) the business income disadvantage associated with institutional stigma.

In Model 8, we interact institutional stigma with both social capital and human capital variables. The observations we made in reference to Models 3 and 7 continue to hold in this model. According to this model, one standard deviation increase in social capital of business owners from non-stigmatized communities is associated with 17.8% higher business income, whereas for Dalit business owners, a one standard deviation increase in social capital is associated with an increase of only 6.2%. One standard deviation increase in human capital is associated with 16.1% increase in business income of both groups. Fig 1 (left) presents the interaction plots (based on Model 8) between institutional stigma and social capital, which highlight that the business owners who primarily benefit from social capital are those who do not face institutional stigma. Fig 1 (right) presents the interaction plots (based on Model 8) between institutional stigma and human capital, which highlights that human capital benefits both types of business owners.

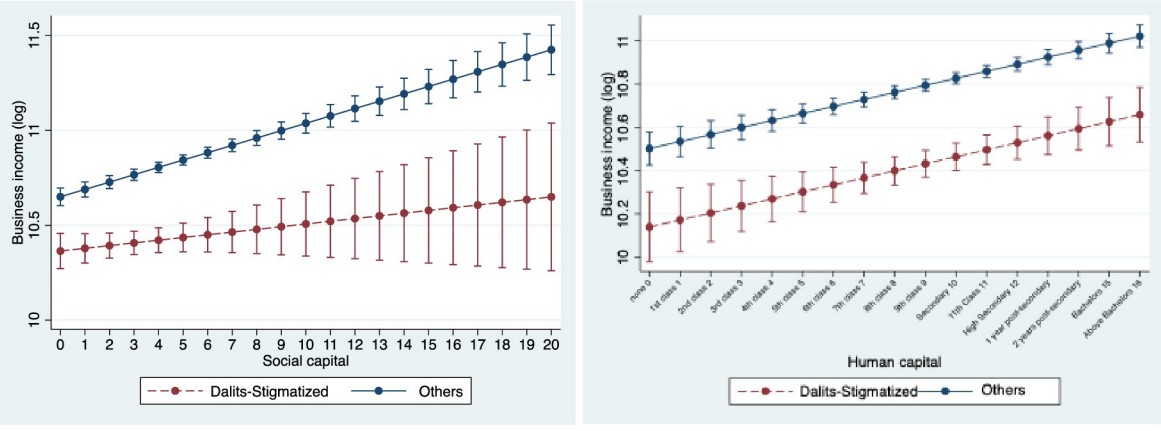

**Fig 1. Plot of the Interaction Between Institutional Stigma and (left) Social Capital and (right) Human Capital (based on Model 8 in Table 1).** *Notes.* The figures display estimates of business income for business owners from communities that face and do not face Institutional stigma. The vertical bars exhibit 95% confidence intervals.

In Model 9 we interact institutional stigma with bridging and bonding social capital, as well as with human capital. The observations we made in Models 6 and 7 continue to hold in this model. According to this model, one standard deviation increase in the bridging social capital of business owners from non-stigmatized communities is associated with 11% higher business income, whereas for business owners who face institutional stigma, it is associated with no increase at all. One standard deviation increase in bonding social capital is associated with an increase in the business incomes of both groups by 8.6%, whereas a one standard deviation in human capital is associated with a 16.18% increase for both groups. Fig 2 (left) presents the interaction plots (based on Model 9) between institutional stigma and bridging social capital highlighting again that business owners who primarily benefit from bridging social capital are those who do not face institutional stigma. Fig 2 (right) presents the interaction plots (based on Model 9) between institutional stigma and bonding social capital highlighting that bonding social capital benefits both types of business owners.

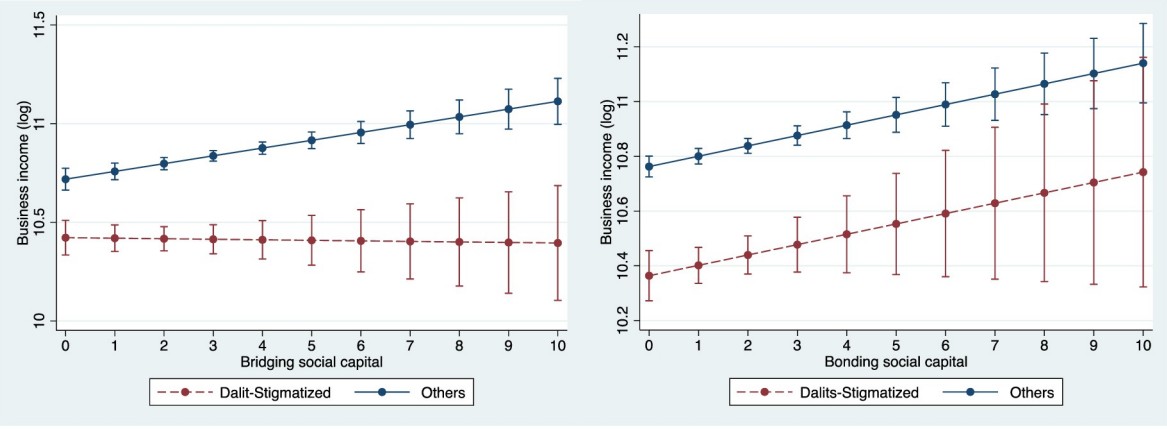

**Fig 2. Plot of the Interaction Between Institutional Stigma and (left) Bridging Social Capital and (right) Bonding Social Capital (based on Model 9 in Table 1).** *Notes.* The figure displays estimates of business income for business owners from communities that face and do not face Institutional stigma. The vertical bars exhibit 95% confidence intervals.

These results provide consistent support for our hypotheses that institutional stigma is negatively associated with business income (H1), and that this negative association is accentuated at higher levels of social capital (H2), especially bridging social capital (H3). On the other hand, according to our analysis, human capital neither attenuates (as hypothesized in H4) nor accentuates (unlike social capital) the negative association between institutional stigma and business income.

## Robustness tests

We also run further robustness tests using a variety of techniques, to check the rigor of our main results–income gap (H1), absence of social capital (H2) and human capital (H4) benefits). First, we replicate Table 1 by removing industry fixed effects to see if industry specific factors may drive our results. Our results remain unchanged. Second, we replicate Table 1 on suitable subsamples: i) a subsample comprising only of business-owning households with disadvantaged background to see whether Dalits, being stigmatized, have an income disadvantage even in a sample that is made up of business-owners from disadvantaged groups only. Our results show that even in this sample Dalits face an income gap of 16.9%; ii) a subsample comprising only of business-owning households in non-metro areas to see whether Dalits have a higher income disadvantage in non-metro urban and rural areas, where stigma might play a larger role. The income gap 15.7% remains similar to the full sample. Similar to our main result, social capital accentuates this income gap, while human capital does not.

Third, there may be genuine concerns whether comparing the entire samples of Dalit business owners to non-Dalit business owners in an OLS model, regardless of the battery of controls, is an apples-to-apples comparison. Hence, we also try to compare Dalit business owners to business owners that exhibit closely matching characteristics. For this, we use a variety of matching techniques, to replicate our income gap result, matching business-owning households on Urban location, Average income of the location, Land, Human capital and Social capital, using (i) Coarsened Exact Matching (CEM) and (ii) Propensity Score matching techniques. We (i) compare business owning Dalit households with business owning households in other groups, (ii) business owning Dalit households with business owning households from other disadvantaged groups, and (iii) non-metro located business owning Dalit households with non-metro located business owning households from other disadvantaged groups. In all the above cases, our results are similar to our main results. Results of these robustness tests are summarized in Table 2.

## Discussion

We found that business owners who face institutional stigma (i.e., Dalits) have a lower income compared to those who do not face such stigma (i.e., non-Dalits), including those who face socioeconomic disadvantage (i.e., OBCs, Adivasis, and Muslims). This income gap is higher at higher levels of social capital, especially bridging social capital, suggesting that institutional stigma inhibits the degree to which stigmatized individuals are able to leverage their social capital, as compared to non-stigmatized individuals. In contrast, human capital, in the form of education, improves the incomes of Dalits to a similar degree as it improves the incomes of non-Dalits.

Our findings should be interpreted in the context of their limitations. The measure of social capital that is available in this survey captures the presence of ties to different groups and professions, but not their number or strength. Therefore, it is a coarse indicator of social capital, but one that presents a reasonable trade-off, given the coverage provided in the survey and also in light of all the other information asked about in this survey (much of which we use to

**Table 2. Robustness tests.** Estimates of Effect Sizes for Model 2 (main effect) and Model 8 (interaction effects) in Table 1.

| | Model 2 | Model 8 | | | Observations |
|---|---|---|---|---|---|
| | Income gap (Dalits) | Social capital (Dalits) | Social capital (others) | Human capital (all) | |
| Based on main results (as presented in Table 1) | -15.89% | 6.24% | 17.84% | 16.13% | 8,506 |
| • No Industry Fixed Effects | -15.21% | 7.23% | 20.06% | 15.70% | 8,506 |
| • Disadvantaged only | -16.89% | 3.29% | 16.22% | 16.20% | 6,021 |
| • non-Metro locations only | -15.72% | 4.04% | 19.21% | 15.97% | 7,694 |
| • Coarsened Exact Matching (CEM) | -18.45% | 7.02% | 17.21% | 16.93% | 8,181 |
| • CEM—Disadvantaged only | -18.70% | 4.54% | 14.69% | 16.08% | 5,828 |
| • CEM—Disadvantaged in non-Metro only | -18.62% | 3.49% | 16.35% | 16.41% | 5,323 |
| • Propensity Score Matching (PScore) | -14.53% | 7.10% | 12.90% | 21.37% | 2,372 |
| • PScore—Disadvantaged only | -17.96% | 3.18% | 21.98% | 18.50% | 2,372 |
| • PScore—Disadvantaged in non-Metro only | -17.72% | 1.92% | 21.08% | 15.17% | 2,084 |

*Note*: Presentation of the magnitude of the effects for Model 8 are calculated on the basis of one standard deviation change in the corresponding variable in that sample. For example, one standard deviation in increase in social capital for Dalits increases their income by 4.07% in a sample that is run on observations that come from non-Metro locations only, whereas this same increase is 19.31% for others. Coarsened Exact Matching (CEM) and Propensity Score Matching (PScore) is conducted by matching observations on variables Urban location, Average income of the location, Land, Human capital and Social capital. CEM values are coarsened in bands between 0th, 25th, 50th, 75th and 100th percentile. PScore matching follows the 1-to-1 no replacement matching with the nearest neighbor.

improve the specification of our model and control for possible confounding variables). We also note that the cross-sectional data limits the extent to which we can make causal inferences, although we control for a number of household, neighbourhood, industry, and regional characteristics. Even though in this case, the institutional stigma indicator we use is *highly* unlikely to change as a result of changes in the outcome we study. Notwithstanding these limitations, our study makes contributions to research on economic inequality, as well as to stigma research, and has implications for management practice and policy.

Management research has a long and robust tradition of examining inequalities rooted in demographic characteristics, such as gender, class, and race [4, 20]. However, management scholars have only recently begun to pay attention to how caste affects economic action in organizational settings [13, 17, 62–66]. We contribute to this emerging theme with a pan-India study of how caste affects income gaps of business owners.

Our results show have important implications for inequality research and policy. The Indian government has developed several initiatives for the mobility of Dalits and other disadvantaged communities, who number a billion in India alone. These initiatives take a similar form for all communities, but our study indicates that these groups (i.e., Dalits on the one hand, who are stigmatized, and other historically disadvantaged–but not similarly stigmatized–groups on the other hand) face different social processes, which need to be considered in designing initiatives to reduce inequalities. Our results can partially explain the limited impact of such traditional initiatives for Dalits and underscore the need to rethink development strategies involving human capital and social capital. Our findings have implications for understanding the sources of inequalities and strategies for reducing them for stigmatized communities (i.e., Dalits) that number 250–300 million in India alone.

Recent discourse is drawing parallels between race and caste to explain the structural nature of inequalities around the world (e.g., the book Caste by Wilkerson [67]). More broadly, research attention is directed towards understanding systems of inequality around the world and learning from their experiences (e.g., the book Capital and Ideology by Piketty [2], which praised the inequality reduction strategies of India). Our study contributes to this emerging

global conversation and can inform strategies being developed to reduce racial inequalities. Our study helps to broaden our understanding of stigma (as we highlight the social and institutional nature of stigma to explain our findings), which has predominantly been understood as arising out of occupations and actions of individuals and businesses.

## Supporting information

**S1 Checklist. Inclusivity in global research.**
(DOCX)

**S1 File.**
(DOCX)

## Author Contributions

**Conceptualization:** Prateek Raj, Thomas J. Roulet, Hari Bapuji.

**Data curation:** Prateek Raj.

**Formal analysis:** Prateek Raj.

**Investigation:** Hari Bapuji.

**Methodology:** Prateek Raj.

**Project administration:** Prateek Raj, Hari Bapuji.

**Supervision:** Hari Bapuji.

**Writing – original draft:** Prateek Raj, Thomas J. Roulet, Hari Bapuji.

**Writing – review & editing:** Prateek Raj, Thomas J. Roulet, Hari Bapuji.

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
