## [Decision Letter · Decision Letter 0]

3 Jun 2024

PONE-D-24-13445It’s not who you know, but who you are: Explaining income gaps of stigmatized-caste business owners in India

PLOS ONE

Dear Dr. Roulet,

Thank you for submitting your manuscript to PLOS ONE. After careful consideration, we feel that it has merit but does not fully meet PLOS ONE’s publication criteria as it currently stands. Therefore, we invite you to submit a revised version of the manuscript that addresses the points raised during the review process. 

Based on my reading of the manuscript, your work is a systematic and critical interdisciplinary enterprise. You may reflect more on the meaning of diffused prejudice and stigmatized identity. I hope the reviewers' comments below will help streamline the manuscript to the criteria of the Journal

We look forward to receiving your revised manuscript.

Kind regards,

Chetan Sinha

Academic Editor

PLOS ONE

“no competing interests”

Reviewers' comments:

Reviewer's Responses to Questions

**Comments to the Author**

1. Is the manuscript technically sound, and do the data support the conclusions?

Reviewer #1: Partly

Reviewer #2: Yes

2. Has the statistical analysis been performed appropriately and rigorously? 

Reviewer #1: Yes

Reviewer #2: I Don't Know

3. Have the authors made all data underlying the findings in their manuscript fully available?

Reviewer #1: Yes

Reviewer #2: Yes

4. Is the manuscript presented in an intelligible fashion and written in standard English?

Reviewer #1: No

Reviewer #2: Yes

5. Review Comments to the Author

Reviewer #1: Mention introduction giving the significance and objective of your study. The first paragraph must be in your own words without citations.

Literature review and methodology section is missing.

Give a structured LR and explain the research methodology clearly.

Mention hypothesis in the methodology section.

Future scope of research is missing.

Reviewer #2: I like this paper and would like to see it in print. However, I have a few issues which you don't have to address here but in future if you choose to pursue it. One, Social Capital (SC) as a framework may not be appropriate in a caste-infested country. Even in America, SC primarily has been a factor among one social group, the WASPs. I maybe wrong but I feel Blacks cannot join a tennis club or a neighborhood volunteer group or a church and later use it as a vehicle for career advancement. So, in a homogeneous group bridging SC is easy. Caste is heterogeneous. SC is a class concept. Two, father's occupation is no predictor of mobility. Defying the Odds by Devesh Kapur et al chronicles how first-generation, (mostly) illiterate and landless Dalits succeeded as entrepreneurs - without hiding their identity. Three, having land (mostly in small pieces) hinders mobility and entrepreneurship as it ties them down to the same village. Four, since entrepreneurship among Dalits is a new phenomenon since late 1990s, you may have compared first generation Dalit entrepreneurs with entrenched groups' who have been in businesses for generations, even if they are disadvantaged social groups. Five, some qualitative stuff adds luster to any work based on numbers. All the five points are beside the point, because I feel your paper will encourage/ provoke other scholars to take it forward.,

6. PLOS authors have the option to publish the peer review history of their article (what does this mean?). If published, this will include your full peer review and any attached files.

Reviewer #1: No

Reviewer #2: No

---

## [Author Response · Author response to Decision Letter 0]

1 Jul 2024

See the attached Response to reviewers' documents

---

## [Editor Report · Decision Letter 1]

10 Jul 2024

It’s not who you know, but who you are: Explaining income gaps of stigmatized-caste business owners in India

PONE-D-24-13445R1

Dear Dr. Roulet,

We’re pleased to inform you that your manuscript has been judged scientifically suitable for publication and will be formally accepted for publication once it meets all outstanding technical requirements.

Kind regards,

Chetan Sinha

Academic Editor

PLOS ONE
---

## [Editor Report · Acceptance letter]

16 Jul 2024

PONE-D-24-13445R1 

PLOS ONE

Dear Dr. Roulet, 

I'm pleased to inform you that your manuscript has been deemed suitable for publication in PLOS ONE. Congratulations! Your manuscript is now being handed over to our production team.

Kind regards, 

on behalf of

Dr. Chetan Sinha 

Academic Editor

PLOS ONE